# A polymorphism in the tumor suppressor p53 affects aging and longevity in mouse models

Yuhan Zhao[1], Lihua Wu[1], Xuetian Yue[1], Cen Zhang[1], Jianming Wang[1], Jun Li[1], Xiaohui Sun[1,2], Yiming Zhu[2], Zhaohui Feng[1,3], Wenwei Hu[1,3]*

[1]Department of Radiation Oncology, Rutgers Cancer Institute of New Jersey, Robert Wood Johnson Medical School, Rutgers, the State University of New Jersey, New Brunswick, United States; [2]Department of Epidemiology and Biostatistics, School of Public Health, Zhejiang University, Hangzhou, China; [3]Department of Pharmacology, Rutgers, the State University of New Jersey, Piscataway, United States

**Abstract** Tumor suppressor p53 prevents early death due to cancer development. However, the role of p53 in aging process and longevity has not been well-established. In humans, single nucleotide polymorphism (SNP) with either arginine (R72) or proline (P72) at codon 72 influences p53 activity; the P72 allele has a weaker p53 activity and function in tumor suppression. Here, employing a mouse model with knock-in of human *TP53* gene carrying codon 72 SNP, we found that despite increased cancer risk, P72 mice that escape tumor development display a longer lifespan than R72 mice. Further, P72 mice have a delayed development of aging-associated phenotypes compared with R72 mice. Mechanistically, P72 mice can better retain the self-renewal function of stem/progenitor cells compared with R72 mice during aging. This study provides direct genetic evidence demonstrating that p53 codon 72 SNP directly impacts aging and longevity, which supports a role of p53 in regulation of longevity.
DOI: https://doi.org/10.7554/eLife.34701.001

*For correspondence: wh221@cinj.rutgers.edu

**Competing interests:** The authors declare that no competing interests exist.

## Introduction

Aging is a complex process of time-dependent series of progressive loss of functions and structures of all systems, which leads to an increased vulnerability to death (*López-Otín et al., 2013*). Cancer is an age-associated disease, which can lead to both premature death and age-associated increase in morbidity and mortality (*Campisi and Yaswen, 2009*). Tumor suppressor p53 plays a pivotal role in tumor prevention (*Feng et al., 2008*; *Vousden and Prives, 2009*). Loss or disruption of p53 function is often a prerequisite for tumor initiation and development. In humans, more than 50% of all human tumors contain mutations in the *p53* gene (*Olivier et al., 2002*). In mice, loss of both *Trp53* alleles (p53-/-) leads to the development of tumors early in life and a reduced lifespan compared with wild type mice (*Donehower et al., 1992*). Therefore, p53 ensures longevity by preventing cancer development early in life.

Longevity depends on the balance between tumor suppression and tissue renewal mechanisms (*Campisi and Yaswen, 2009*). While genomic instability is a hallmark of aging, stem cell exhaustion is another important hallmark of aging (*López-Otín et al., 2013*). It has been indicated that the anti-proliferative function of p53 which is crucial for tumor suppression could affect self-renewal function of stem/progenitor cells and contribute to aging (*van Heemst et al., 2005*; *Donehower, 2002*). However, the precise role of p53 in aging process and longevity has not been clearly established. Inconsistent results on aging and longevity have been reported in different mouse models in which

**eLife digest** How long most animals live depends on the balance between the biological processes that allow them to regenerate their tissues when damaged and those that prevent them from developing cancer. Regeneration relies mostly on cells, in particular stem cells, dividing to make new cells, while cancer occurs when cell division becomes uncontrolled.

Tumor suppressor genes protect against cancer. One such gene encodes a protein called p53 that eliminates damaged cells before they can become cancerous. The p53 protein is also believed to be involved in regulating how quickly an animal ages and how long it lives, but this second role has not yet been clearly established.

Previous studies using different strategies to change the activity of p53 in several mouse models have led to inconsistent results. However, the mouse models used in these earlier studies did not reflect how p53 works under normal conditions.

Zhao et al. have now used mice in which the mouse gene for p53 was replaced with one of two versions of the equivalent human gene to study its impact on lifespan and the aging process. The two versions of p53 only differ slightly; a single building block of the protein, the amino acid at position 72, is a proline in one version but an arginine in the other. This difference makes one version of p53 weaker than the other; in other words, it is less able to eliminate damaged cells. Zhao et al. revealed that the mice with the weaker p53 lived for longer and appeared to age more slowly too. Further experiments showed that the stem cells in the mice with a weaker p53 were able to keep dividing and create new cells for longer. This is important because a decline in this activity – which is known as self-renewal – is a hallmark of aging.

Together these findings show that a small yet common change in p53 impacts both aging and lifespan, possibly by altering how stem cells are regulated. Further work is now needed to better understand why the different versions of p53 have different effects on stem cells.

DOI: https://doi.org/10.7554/eLife.34701.002

the p53 activity has been manipulated through different strategies (*Tyner et al., 2002*; *Dumble et al., 2007*; *Maier et al., 2004*; *Liu et al., 2010*; *García-Cao et al., 2002*; *Mendrysa et al., 2006*; *Matheu et al., 2007*). Specifically, transgenic mice with constitutively elevated p53 activity by expression of certain p53 mutants or a short p53 isoform showed increased cancer resistance but premature aging phenotypes (*Tyner et al., 2002*; *Dumble et al., 2007*; *Maier et al., 2004*; *Liu et al., 2010*). The 'super p53' mice with a regulated hyperactive p53 activity by having an extra copy of the wild type *Trp53* gene were resistant to cancer but did not exhibit signs of accelerated aging (*García-Cao et al., 2002*; *Mendrysa et al., 2006*). Interestingly, the 'super p53' mice with an extra copy of *Ink4/Arf* showed extended longevity (*Matheu et al., 2007*; *Matheu et al., 2009*). It is worth noting that mouse models used in these studies did not reflect p53 activation under physiological conditions. It is therefore critical to address the role of p53 in the aging process and longevity using a proper mouse model reflecting the p53 activity under physiological conditions.

*TP53* is a haplo-insufficient gene, a little decrease in p53 levels or activity (e.g. 2-fold difference) significantly impacts tumorigenesis (*Venkatachalam et al., 2001*; *Berger and Pandolfi, 2011*; *Bond et al., 2004*). p53 protein levels and activity are under tight regulation in cells (*Feng et al., 2008*; *Vousden and Prives, 2009*). In humans, naturally occurring single nucleotide polymorphisms (SNPs) in the p53 pathway, which modulate the activity or levels of p53, have been found to significantly impact cancer risk (*Bond et al., 2004*; *Whibley et al., 2009*; *Lin et al., 2008*; *Basu and Murphy, 2016*). p53 codon 72 SNP is a common coding SNP in the *TP53* gene, which results in either an arginine (R72) or a proline (P72) residue at codon 72. We and others have reported that compared with the R72 allele, the P72 allele displays a weaker p53 transcriptional activity towards a group of its target genes, many of which are involved in apoptosis and suppressing cell transformation (*Dumont et al., 2003*; *Jeong et al., 2010*). Studies in human populations indicate that p53 codon 72 SNP may modify cancer risk, but currently the consensus has not been reached on this in the literature (*van Heemst et al., 2005*; *Whibley et al., 2009*). Several studies of aged or general human populations indicate that the P72 carriers have an increased lifespan despite an increased mortality from cancer (*van Heemst et al., 2005*; *Bojesen and Nordestgaard, 2008*; *Smetannikova et al.,*

*2004*). These epidemiological results support the dual functions of p53 in longevity, and suggest that codon 72 SNP may have an impact upon aging and longevity. Considering the genetic background variations of human populations and environmental factors in epidemiological studies, the precise role of p53 codon 72 SNP in aging and longevity remains elusive.

In this study, we employed a mouse model with knock-in of human *TP53* gene (Hupki) carrying codon 72 SNP to directly investigate the impact of p53 codon 72 SNP upon longevity and its underlying mechanism. The Hupki mice carrying codon 72 SNP recapture the impacts of codon 72 SNP upon p53 transcriptional activity and function in tumor suppression, which is widely used for studies on p53 and codon 72 SNP (*Feng et al., 2011*; *Kung et al., 2016*; *Azzam et al., 2011*; *Reinbold et al., 2008*; *Frank et al., 2011*; *Leu et al., 2013*). We found that despite the increased cancer risk, P72 mice that have escaped tumor development have a longer lifespan than R72 mice and display a delay of age-associated phenotypes compared with R72 mice. Mechanistically, P72 mice have a better ability to retain the self-renewal function of stem/progenitor cells compared with R72 mice during the aging process. Long-term stem cells from aging P72 mice have better engraftment and repopulation abilities than aging R72 mice. In turn, P72 mice have less expansion of long-term stem/progenitor cells than R72 mice during the aging process. Taken together, our study provides direct genetic evidence demonstrating that human p53 codon 72 SNP has a direct impact upon aging and longevity in vivo, which supports the role of p53 in longevity.

## Results

### The lifespans of mice carrying human p53 Codon 72 SNP

To investigate the impact of human p53 codon 72 SNP upon aging and the lifespan, Hupki mice with knock-in of human *TP53* gene carrying codon 72 SNP in place of the corresponding mouse *Trp53* gene were employed (*Kung et al., 2016*; *Frank et al., 2011*; *Leu et al., 2013*). It has been reported that p53 protein levels in different tissues are comparable between R72 and P72 mice, which was confirmed in this study (*Figure 1—figure supplement 1A*) (*Kung et al., 2016*; *Frank et al., 2011*; *Leu et al., 2013*). Previous studies including ours showed that the P72 allele in these mice has a weaker transcriptional activity towards a subset of p53 target genes than the R72 allele, suggesting that these mice retain the function of p53 codon 72 SNP in human (*Feng et al., 2011*; *Kung et al., 2016*; *Azzam et al., 2011*). Because the lifespan of mice varies among different inbred strains, Hupki mice with p53 codon 72 SNP were backcrossed to mice with different genetic backgrounds, including 129SV[sl] and C57BL/6J, for ten generations to establish p53 codon 72 SNP Hupki mice in 129SV[sl] and C57BL/6J backgrounds, respectively. The lifespan of mice with p53 codon 72 SNP in 129SV[sl] and C57BL/6J backgrounds was measured in a cohort of ~150 mice for each genotype. The median survival age was 740 days in 129SV[sl] mice and 490 days in C57BL/6J mice, respectively (*Figure 1—figure supplement 1B and C*), which is consistent with previously reported lifespans of these two mouse strains (*Storer, 1966*).

In 129SV[sl] mice, P72 mice showed an overall longer lifespan compared with R72 mice; the median survival age was 759 days for P72 mice and 697 days for R72 mice, respectively (Log-rank test: p<0.0001) (*Figure 1A*). The causes of death included tumor, inflammation (including dermatitis), ocular lesion, urinary syndrome, nephropathy, etc., which are common causes of death in 129SV[sl] mice as reported by previous studies (*Marx et al., 2013*; *Brayton et al., 2012*; *Radaelli et al., 2016*) (*Table 1*). For those mice died from non-neoplastic events, P72 mice showed a significantly longer lifespan than R72 mice; the median survival was 768 days for P72 mice and 673 days for R72 mice, respectively (Log-rank test: p<0.0001) (*Figure 1B*). For those mice died from neoplastic diseases, R72 mice (with a median survival of 774 days) showed a longer lifespan than P72 mice (with a median survival of 756 days) (Log-rank test: p=0.015) (*Figure 1C*). Further analysis of mice older than 18 months, which are equivalent to humans older than 60 years (*Dutta and Sengupta, 2016*), showed that P72 mice had a longer lifespan (with a median survival of 780 days) than R72 mice (with a median survival of 715 days) (Log-rank test: p<0.0001) (*Figure 1D*).

Similar results were observed in C57BL/6J mice. P72 mice had an overall longer lifespan (with a median survival of 495.5 days) than R72 mice (with a median survival of 481 days) (Log-rank test: p=0.015) (*Figure 2A*). For those mice died from non-neoplastic events, P72 mice had a significantly longer lifespan (with a median survival of 564.5 days) than R72 mice (with a median survival of 438

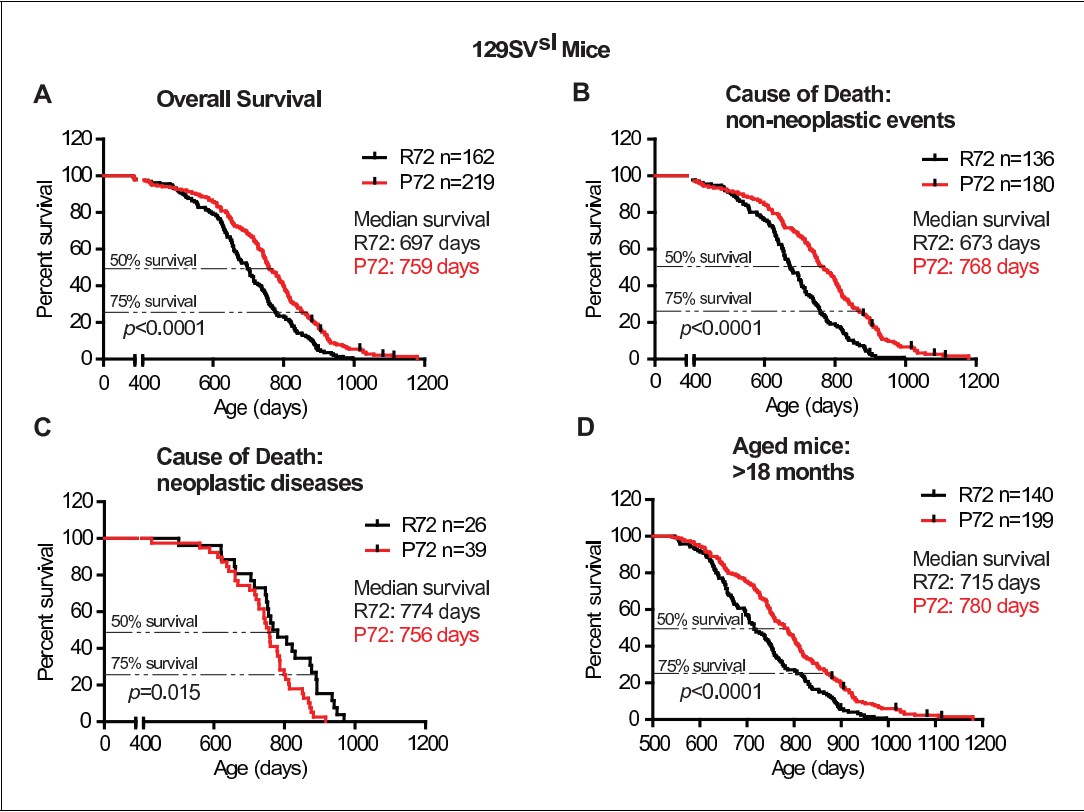

**Figure 1.** The lifespans of 129SV[sl] p53 R72 mice and p53 P72 mice. (**A**) The overall survival of 129SV[sl] R72 mice (n = 162) and P72 mice (n = 219). The median survival is 697 days and 759 days for R72 and P72 mice, respectively. p<0.0001. (**B**) Survival of 129SV[sl] R72 mice (n = 136) and P72 mice (n = 180) died from non-neoplastic events. The median survival is 673 days and 768 days for R72 and P72 mice, respectively. p<0.0001. (**C**) Survival of 129SV[sl] R72 mice (n = 26) and P72 mice (n = 39) died from neoplastic diseases. The median survival is 774 days and 756 days for R72 mice and P72 mice, respectively. p=0.015. (**D**) Survival of 129SV[sl] R72 mice (n = 140) and P72 mice (n = 199) older than 18 months. The median survival is 715 days and 780 days for R72 mice and P72 mice, respectively. p<0.0001. Log-rank test was used to analyze the difference in survival.

DOI: https://doi.org/10.7554/eLife.34701.003

The following figure supplement is available for figure 1:

**Figure supplement 1.** The p53 protein levels in the bone marrow from 129SV[sl] p53 Hupki mice and the lifespans of 129SV[sl] and C57BL/6J p53 Hupki mice.

DOI: https://doi.org/10.7554/eLife.34701.004

days) (Log-rank test: p<0.0001) (*Figure 2B*). For those mice died from neoplastic diseases, R72 mice (with a median survival of 566 days) had a longer lifespan than P72 mice (with a median survival of 411 days) (Log-rank test: p=0.0084) (*Figure 2C*). Further analysis of mice older than 18 months showed that P72 mice had a longer lifespan (with a median survival of 693 days) than R72 mice (with a median survival of 657 days) (Log-rank test: p<0.0001) (*Figure 2D*).

## P72 mice showed delayed aging-associated phenotypes

Our results that mice carrying different p53 codon 72 SNP have different lifespans suggest that p53 codon 72 SNP impacts the aging process. Therefore, several aging-associated phenotypes were examined in R72 and P72 mice at different ages.

During the aging process, mice develop lordokyphosis which is characterized by an increased curvature of the spine (*López-Otín et al., 2013*). In this study, skeleton structures of 129SV[sl] mice at different ages were imaged and reconstructed by a micro-CT scan. A narrowing of the spine angle indicates an increase in lordokyphosis. In 6-month-old mice, lordokyphosis was not observed, and there was no significant difference in spinal curves between R72 and P72 mice (*Figure 3A&B*). In 18-month-old mice, lordokyphosis was observed. Notably, 18-month-old R72 mice developed more

**Table 1.** Major contributing causes of death in mice evaluated at end of life.

| | 129SV[sl] | | C57BL/6J | |
|---|---|---|---|---|
| | R72 | P72 | R72 | P72 |
| | n (%) | n (%) | n (%) | n (%) |
| Neoplasm | 26 (16.3) | 39 (18.3) | 48 (32.9) | 56 (35) |
| Nonspecific systemic disease | 33 (20.6) | 47 (22.1) | 21 (14.4) | 31 (19.4) |
| Ocular lesion | 38 (23.8) | 53 (24.9) | 7 (4.8) | 5 (3.1) |
| Dermatitis | 5 (3.1) | 4 (1.9) | 34 (23.3) | 43 (26.9) |
| Megaesophagus | 15 (9.4) | 24 (11.3) | | |
| Urinary syndrome/nephropathy | 18 (11.3) | 21 (9.9) | 9 (6.2) | 4 (2.5) |
| Neurologic diseases | 5 (3.1) | 7 (3.3) | 4 (2.7) | 3 (1.9) |
| Others | 9 (5.6) | 8 (3.8) | 10 (6.8) | 8 (5.0) |
| Unknown reason | 11 (6.9) | 10 (4.7) | 13 (8.9) | 10 (6.3) |

Note:

Nonspecific systemic disease: age-related or -induced lesions including cardiac and respiratory failure; sepsis and DIC; female reproductive diseases or male urogenital diseases, etc.

Ocular lesion: includes corneal ulceration and chronic keratitis

Neurologic diseases: include head tilt, paresis, paralysis and ataxia

Others: include rectal prolapse, gastrointestinal bleedings

DOI: https://doi.org/10.7554/eLife.34701.005

pronounced lordokyphosis compared with age-matched P72 mice (*Figure 3B*). A similar phenotype was observed in C57BL/6J mice (*Figure 3G&H*).

Another aging-related phenotype in both humans and mice is osteoporosis (*López-Otín et al., 2013*). The mouse tibias bone structure and density were examined by a micro-CT scan followed by 3D reconstruction. The structure and density of the tibias bone between 6-month-old 129SV[sl] R72 and P72 mice were morphologically identical, and showed no sign of osteoporosis (*Figure 3C*). Osteoporosis was observed in both R72 and P72 mice at the age of 18 months. Notably, R72 mice displayed a more obvious sign of osteoporosis than P72 mice (*Figure 3C*). Analysis of tibias bone structure and density of mice at different ages showed aging-related changes, including decreased bone volume/total volume (BV/TV), decreased trabecular number and increased trabecular spacing during aging (*Figure 3D–F*). P72 mice showed a delayed development of all these aging-related changes compared with R72 mice, with the most obvious differences observed at the age of 18 months (*Figure 3D–F*). Similar results were obtained in C57BL/6J mice (*Figure 3I–K*).

Decreases in the skin dermal thickness and subcutaneous adipose tissues occur during the aging process (*López-Otín et al., 2013*). Older 129SV[sl] mice (12–18 month-old) had a thinner dermal layer and less subcutaneous adipose tissues than young mice (6-month-old) (*Figure 4A–C*). There was no obvious difference in the dermal thickness and the amount of subcutaneous adipose tissues between young R72 and P72 mice. Notably, in older mice, R72 mice showed more significant decreases in both skin dermal thickness and subcutaneous adipose thickness compared with P72 mice (*Figure 4A–C*).

One of the hallmarks of aging is the stem cell exhaustion, which leads to the reduced ability of tissue repair (*López-Otín et al., 2013*). Under stress, such as skin wounds, epidermal stem cells exhibit a highly organized and complex self-renewal process to restore the integrity and function of the skin. This ability dampens down as both humans and mice age (*López-Otín et al., 2013*). Therefore, the cutaneous repair ability of mice was examined by measuring the wound healing process which reflects the function of the skin stem cell (*Shaw and Martin, 2009*). Three-mm wounds were introduced in the mouse skin by punch and the wound diameters were measured daily. Both R72 and P72 129SV[sl] young mice (6-month-old) showed an efficient wound healing ability (*Figure 4D*). However, 12- and 18-month-old R72 mice showed a more pronounced decrease in the wound healing ability than age-matched P72 mice (*Figure 4D*). Similar results were obtained in C57BL/6J mice

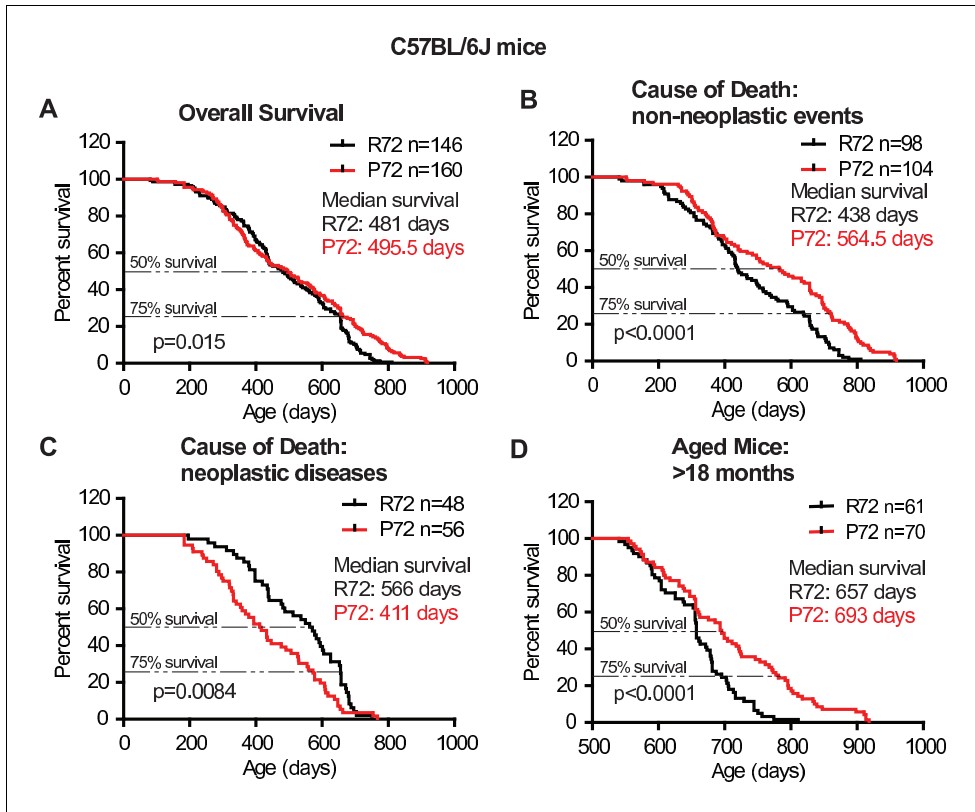

**Figure 2.** The lifespans of C57BL/6J p53 R72 mice and p53 P72 mice. (**A**) The overall survival of C57BL/6J R72 mice (n = 146) and P72 mice (n = 160). The median survival is 481 days and 495.5 days for R72 and P72 mice, respectively. p=0.015. (**B**) Survival of C57BL/6J R72 mice (n = 98) and P72 mice (n = 104) died from non-neoplastic events. The median survival is 438 days and 564.5 days for R72 and P72 mice, respectively. p<0.0001. (**C**) Survival of C57BL/6J R72 mice (n = 48) and P72 mice (n = 56) died from neoplastic disease. The median survival is 566 days and 411 days for R72 mice and P72 mice, respectively. p=0.0084. (**D**) Survival of C57BL/6J R72 mice (n = 61) and P72 mice (n = 70) older than 18 months. The median survival is 657 days and 693 days for R72 mice and P72 mice, respectively. p<0.0001. Log-rank test was used to analyze the difference in survival.
DOI: https://doi.org/10.7554/eLife.34701.006

(*Figure 4E*). These results demonstrate that P72 mice exhibited a delayed aging process compared with R72 mice.

## The impact of p53 Codon 72 SNP upon hematopoietic stem cell self-renewal abilities during the aging process

Stem cell exhaustion is considered as a hallmark of the aging process (*López-Otín et al., 2013*). During aging in both humans and mice, the regeneration ability of stem cells gradually diminishes (*López-Otín et al., 2013*). Ample studies on stem cell aging process have focused on hematopoietic stem cell (HSC) (*Chambers and Goodell, 2007*; *Seita and Weissman, 2010*). Studies using mouse models demonstrated a HSC aging phenotype with the characteristic of the increase of the pool of stem/progenitor cells and the reduction of their self-renewal abilities during the aging process (*Dumble et al., 2007*; *Chambers et al., 2007*). p53 has been indicated to play a critical role in regulating the function of stem/progenitor cells (*Dumble et al., 2007*; *Kaiser and Attardi, 2018*). Here, we investigated the impact of p53 codon 72 SNP upon HSC pool size and self-renewal function during aging.

To this end, the numbers of long term-HSCs (LT-HSCs) as well as proliferating HSCs, which represent HSC pool size and self-renewal function, respectively, were measured in R72 and P72 mice at different ages. Bone marrow cells were isolated from mouse hind limb bones and stained with mature hematopoietic lineage markers. The numbers of LT-HSCs (Lin$^{-/low}$, Sca1$^+$, c-kit$^+$ and CD34$^-$,

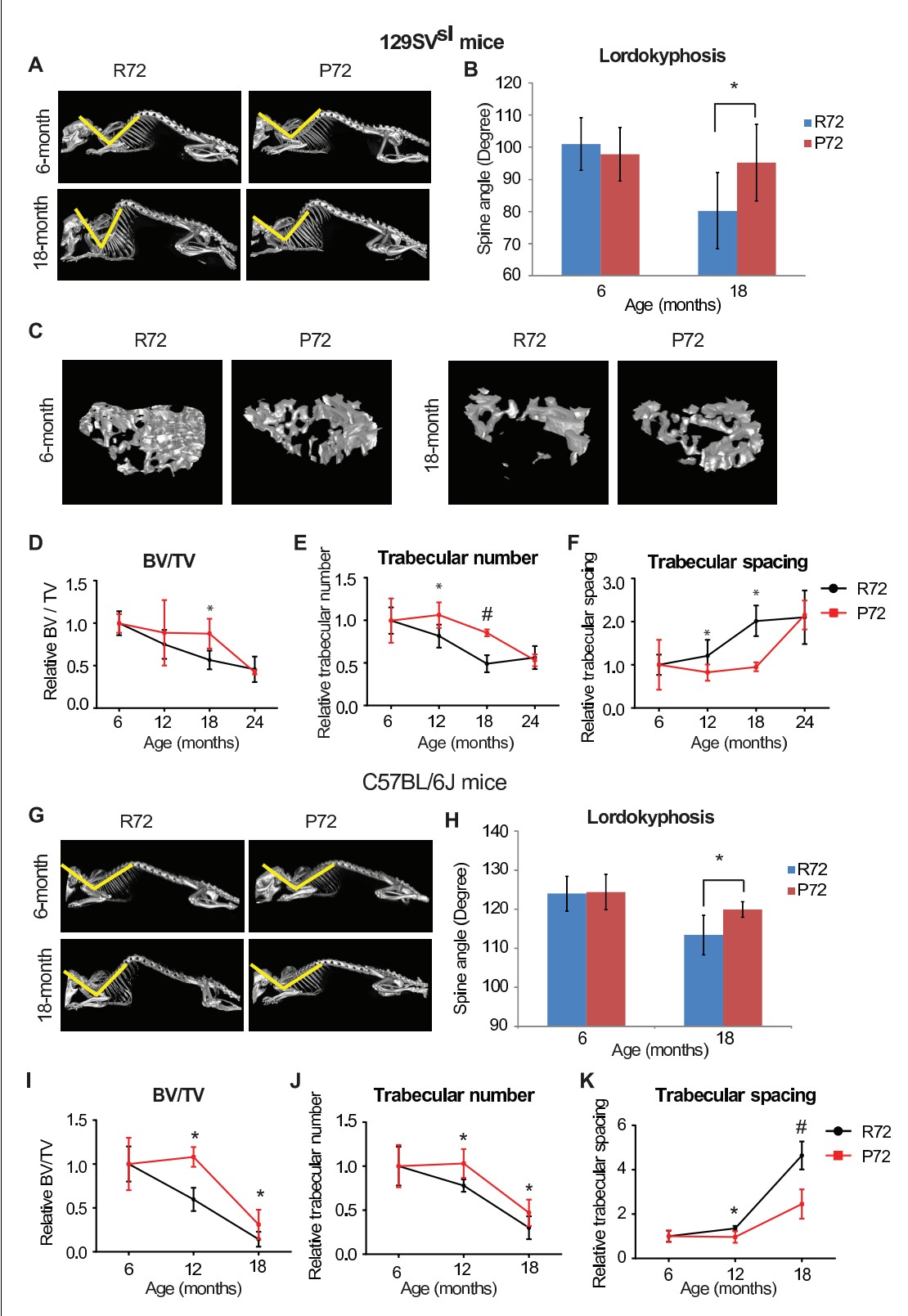

**Figure 3.** p53 P72 mice have a delay in developing aging-associated bone structure phenotypes compared with p53 R72 mice. (**A**) Lordokyphosis in 129SV^sl R72 and P72 mice. Shown are representative images taken with the INVEON PET/CT system of mouse whole skeleton at different ages. (**B**) Average of spine angles from 6-month-old and 18-month-old 129SV^sl R72 and P72 mice. A narrowing of the angle indicates an increase in lordokyphosis. n = 5/group, *: p<0.05; Student's *t*-test. (**C**) Representative micro-CT bone structure images taken with the INVEON PET/CT system of
*Figure 3 continued on next page*

*Figure 3 continued*

tibias from 129SV$^{sl}$ R72 and P72 mice at different ages. (D–F) Quantification of bone volume relative to total volume (D), trabecular number (E) and trabecular spacing (F) from micro-CT scans of tibias from 129SV$^{sl}$ R72 and P72 mice at different ages using INVEON Research Workplace software. Data were presented as mean ± SD. n = 3, *: $p<0.05$, #: $p<0.01$; Student's *t*-test. (G) Lordokyphosis in C57BL/6J R72 and P72 mice. Shown are representative images of mouse whole skeleton at different ages. (H) Average of spine angles from 6-month-old and 18-month-old C57BL/6J R72 or P72 mice. n = 5, *: $p<0.05$; Student's *t*-test. (I–K) Quantification of bone volume relative to total volume (I), trabecular number (J) and trabecular spacing (K) from micro-CT scans of tibias from C57BL/6J R72 and P72 mice at different ages using INVEON Research Workplace software. Data were presented as mean ± SD. n = 3, *: $p<0.05$, #: $p<0.01$; Student's *t*-test.

DOI: https://doi.org/10.7554/eLife.34701.007

The following source data is available for figure 3:

**Source data 1.** Average of spine angle from 6-month-old and 18-month-old R72 and P72 mice.

DOI: https://doi.org/10.7554/eLife.34701.008

Flk2⁻) were determined by FCM analysis (*Figure 5—figure supplement 1*). Consistent with previous reports (*Akunuru et al., 2016*; *Morrison et al., 1996*), the percentage of LT-HSCs in bone marrow cells clearly increased during the aging process in both 129SV$^{sl}$ and C57BL/6J mice (*Figure 5A–C*). R72 mice showed a more rapid increase in the numbers of LT-HSCs than P72 mice during aging. While there was no significant difference in LT-HSC numbers between young 129SV$^{sl}$ R72 and P72 mice, much higher LT-HSC numbers were observed in R72 mice than P72 mice at the age of both 12 and 18 months (*Figure 5A&B*). Similar results were obtained in C57BL/6J mice (*Figure 5C*). These results demonstrated that P72 mice showed a delayed HSC expansion during aging.

To determine the population of functional/proliferating HSCs in R72 and P72 mice at different ages, BrdU-labeled proliferating HSCs were quantified by FCM analysis. As shown in *Figure 5D&E*, the number of proliferating HSCs decreased during aging in both 129SV$^{sl}$ and C57BL/6J mice, which is consistent with previous reports (*Dumble et al., 2007*; *Chambers and Goodell, 2007*). Notably, the decrease of proliferating HSC numbers was more rapid in R72 mice than P72 mice during aging. In 129SV$^{sl}$ mice, the percentage of proliferating HSCs in all HSCs in R72 mice decreased from 45% at the age of 6 months to 28% at the age of 22 months, whereas the decrease of proliferating HSCs in P72 mice was less pronounced: from 44% at the age of 6 months to 34% at the age of 22 months (*Figure 5D*). Similar results were obtained in C57BL/6J mice; the decrease of proliferation HSCs was more rapid in R72 mice than P72 mice during aging (*Figure 5E*).

To further evaluate the self-renewal and repopulation function of HSCs in mice with p53 codon 72 SNP during aging, the engraftment and repopulation abilities of HSCs of mice were determined by bone marrow transplantation assays. Bone marrow cells isolated from donor CD45.2 mice with different p53 codon 72 SNP at different ages were transplanted into lethally irradiated recipient CD45.1 mice along with CD45.1 bone marrow cells (*Figure 6A*). The long-term HSC engraftment and repopulation abilities were evaluated by analyzing CD45.1 or CD45.2 cell surface markers of peripheral blood cells at 16 weeks after transplantation. As shown in *Figure 6B*, while bone marrow cells from 6-month-old R72 and P72 mice showed similar abilities in engraftment and contribution to mature peripheral lymphocytes, bone marrow cells from 18-month-old P72 mice showed a significantly higher engraftment ability than R72 mice. For donors from 129SV$^{sl}$ mice, ~76% *vs.*~68% of lymphocytes were derived from 18-month-old P72 and R72 donors, respectively. Similar results were obtained in C57BL/6J mice; ~74% *vs* ~56% of lymphocytes were derived from 18-month-old P72 and R72 donors, respectively (*Figure 6B*). Taken together, these results demonstrated that P72 mice displayed a delayed aging process in HSC number and function compared with R72 mice, which contributes to the delayed aging phenotypes in P72 mice.

In response to stress, p53 transcriptionally regulates a group of target genes that can lead to different cell fates through inducing growth arrest or apoptosis, etc. Here, 129SV$^{sl}$ mice were employed to examine whether p53 codon 72 SNP differentially regulates the basal expression of its target genes involved in cell cycle arrest (p21) and apoptosis (Puma and Noxa), which in turn impacts the number and function of stem cells. As shown in *Figure 6C*, p21 mRNA expression levels in the bone marrow were slightly higher in P72 mice compared with R72 mice as determined by real-time PCR assays with this difference being more obvious in older mice than young mice. This difference in p21 expression levels was confirmed at the protein level as determined by Western-blot assays (*Figure 6C*). In contrast, the bone marrow from P72 mice displayed slightly lower expression levels

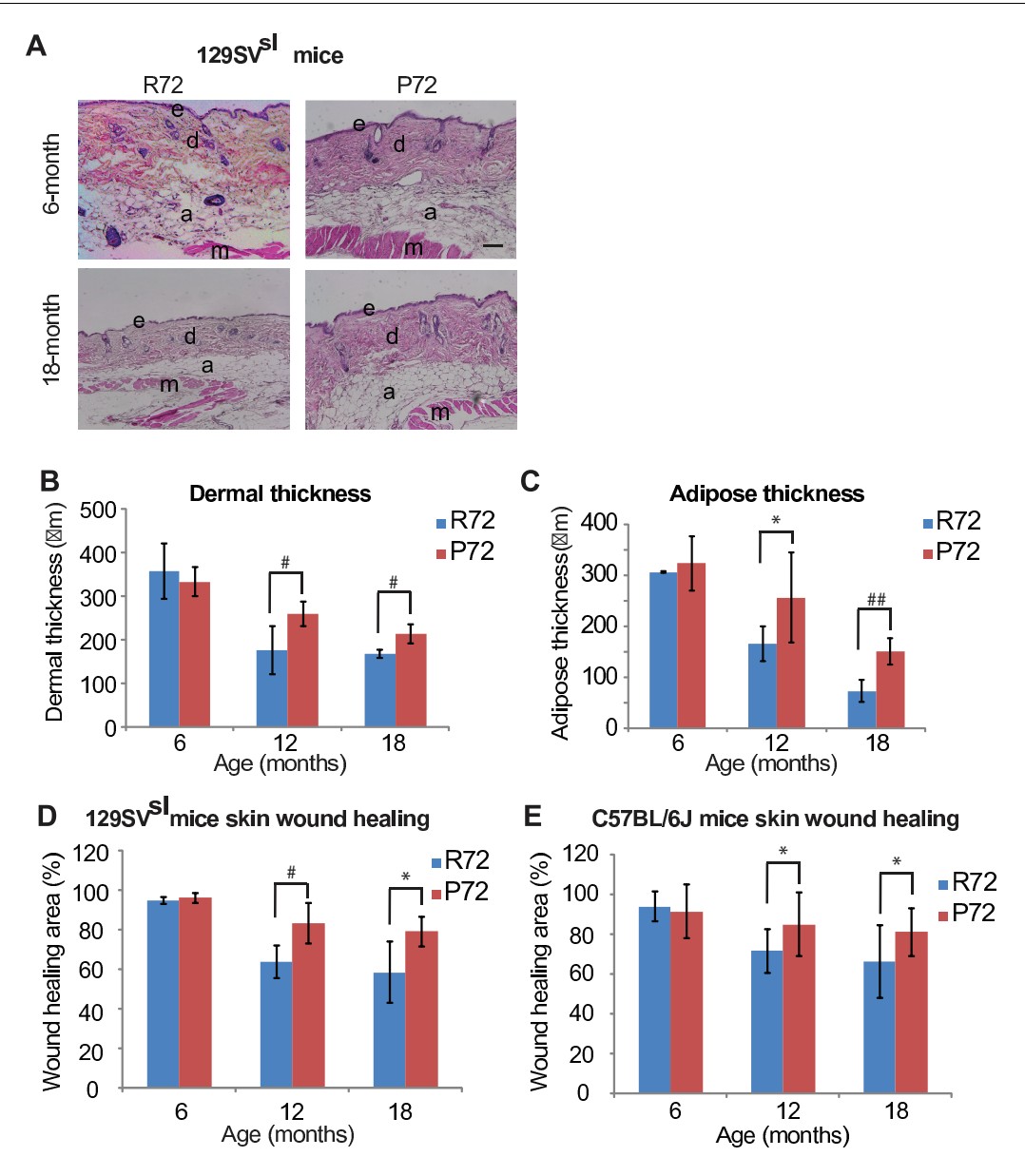

**Figure 4.** p53 P72 mice have a delay in developing aging-associated skin phenotypes compared with p53 R72 mice. (A) H and E staining of cross-sections of dorsal skins from 129SV[sl] R72 mice and P72 mice at different ages. e: epidermis; d: dermis; a: adipose; m: muscle. Scale bar: 100 μm. (B and C) Quantification of dermal thickness (B) and subcutaneous adipose layer thickness (C) of 129SV[sl] R72 mice and P72 mice at different ages. (D and E) Skin wound healing abilities in 129SV[sl] (D) and C57BL/6J (E) R72 and P72 mice at different ages. The skin wound area was quantified as $0.25 \times \pi \times$ width $\times$ length. For (B–E), data were presented as mean ± SD, n ≥ 5/group, *: p<0.05, #:p<0.01, ##: p<0.001; Student's t-test.
DOI: https://doi.org/10.7554/eLife.34701.009
The following source data is available for figure 4:

**Source data 1.** Quantification of dermal thickness of 129SV R72 and P72 mice at different ages.
DOI: https://doi.org/10.7554/eLife.34701.010

of *Puma* and *Noxa* than that from R72 mice (*Figure 6C*). These results demonstrate the differential regulation of the basal expression levels of *p21*, *Puma* and *Noxa* by p53 codon 72 SNP in the bone

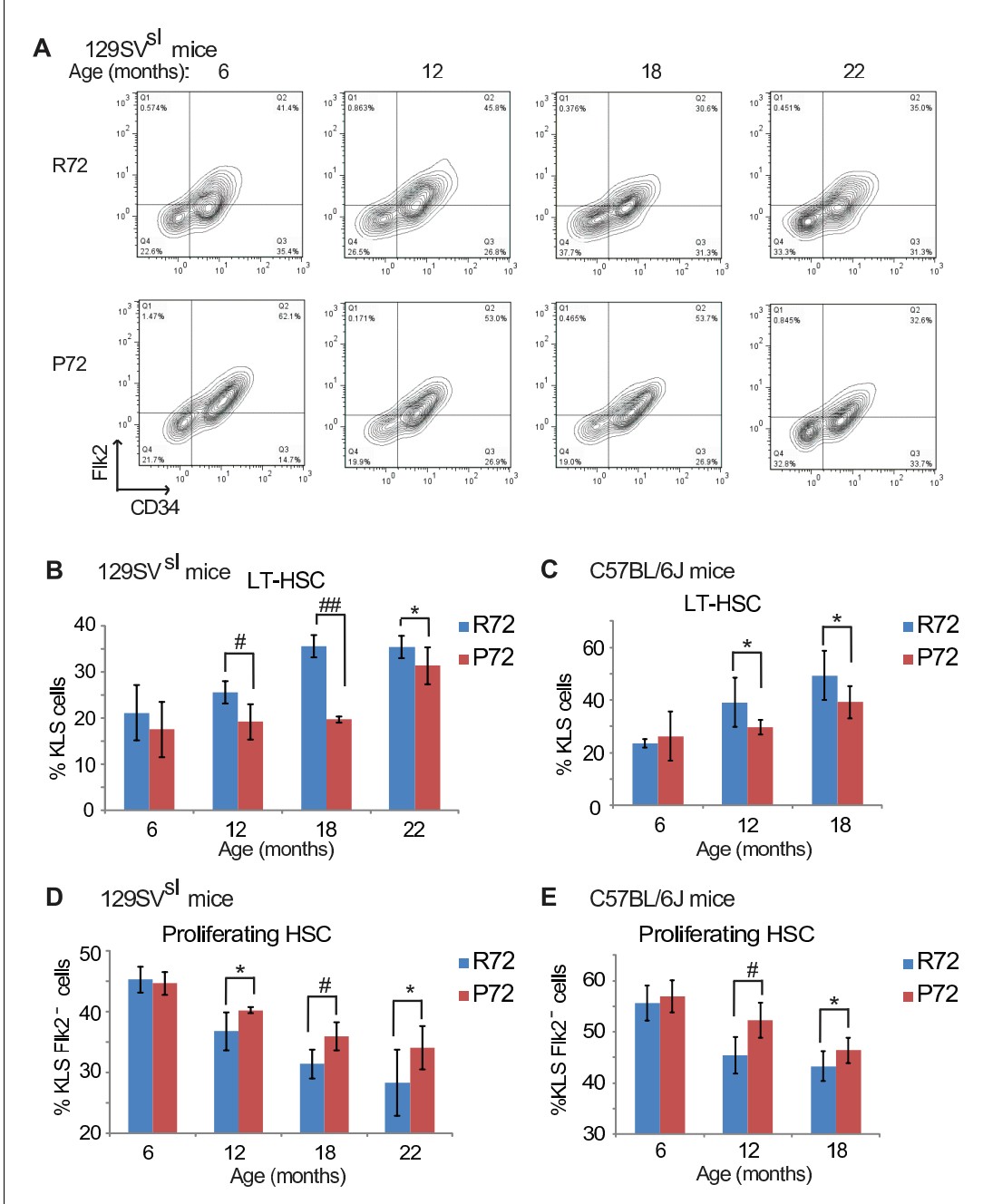

**Figure 5.** HSCs from p53 P72 mice exhibit a delayed aging phenotype compared with HSCs from p53 R72 mice. (A) The representative gating of long-term HSCs (LT-HSCs) in 129SV[sl] R72 mice and P72 mice. LT-HSCs were identified as Lin[-/low], Sca1[+], c-kit[+] and CD34[-], Flk2[-] cells from bone marrow by flow cytometry analysis. (B and C) The bar graph represents the percentage of LT-HSCs in the bone marrow of 129SV[sl] (B) and C57BL/6J (C) R72 mice and P72 mice at different ages. Data were presented as mean ± SD. n ≥ 5/group, *: p<0.05, #: p<0.01, ##: p<0.001; Student's *t*-test. (D and E) The bar graph represents the percentage of proliferating HSCs in the bone marrow of 129SV[sl] (D) and C57BL/6J (E) R72 mice and P72 mice at different ages. Proliferating HSCs were labeled with BrdU for 16 hr in vivo and then identified as Lin[-/low], Sca1[+], c-kit[+], Flk2[-] and BrdU[+] by flow cytometry analysis. Data were presented as mean ± SD. n = 5/group, *: p<0.05, #: p<0.01; Student's *t*-test.

DOI: https://doi.org/10.7554/eLife.34701.011

The following source data and figure supplement are available for figure 5:

**Source data 1.** The percentage of LT-HSCs in the bone marrow of R72 and P72 mice at different ages.
DOI: https://doi.org/10.7554/eLife.34701.013

**Figure supplement 1.** The gating strategy to identify LT-HSCs.

*Figure 5 continued on next page*

*Figure 5 continued*

DOI: https://doi.org/10.7554/eLife.34701.012

marrow, which may contribute to the delayed aging process in HSC number and function observed in P72 mice.

## Discussion

While the role of p53 in assuring longevity through prevention of early cancer development has been well established, its role in regulating aging and longevity aside from cancer prevention has not been well established. Divergent results have been obtained from different mouse models in which the p53 activity was manipulated through different strategies. The increased p53 activity was reported to lead to accelerated aging in some mouse models, but do not affect the lifespan or even prolong the life span in other mouse models (*Tyner et al., 2002*; *Dumble et al., 2007*; *Maier et al., 2004*; *Liu et al., 2010*; *García-Cao et al., 2002*; *Mendrysa et al., 2006*; *Matheu et al., 2007*). These results indicate that p53 can be pro-aging or pro-longevity depending on the context of its regulation and activity. The precise role of p53 in intrinsic aging process, especially under the physiological condition, remains unclear.

Longevity depends on a balance between tumor suppression and tissue renewal mechanisms (*López-Otín et al., 2013*; *Campisi, 2003a*). Declines in stem cells self-renewal and differentiation are critical components of aging (*Campisi and Yaswen, 2009*). The anti-proliferative function of p53, which is crucial for suppression of cancer cells, plays a crucial role in eliminating damaged cells including stem cells (*TeKippe et al., 2003*; *Shounan et al., 1996*). The pleiotropic antagonism theory suggests that certain cellular processes that provide beneficial effects in youth, may compromise organismal fitness later in life (*Campisi, 2003b*). Currently, it is unclear whether p53 has the antagonistic pleiotropy and how the balance of p53 for tumor surveillance and stem cell regulation is regulated.

In humans, functional SNPs have been identified in both p53 and its signaling pathway, such as p53 codon 72 SNP and SNP309 in p53 negative regulator MDM2. These SNPs alter the levels and/ or function of p53. Some of these SNPs, including p53 codon 72 SNP, appear to have undergone the natural selection, which suggests that p53 has evolutionarily-conserved functions other than tumor suppression (*Atwal et al., 2007*; *Atwal et al., 2009*). It is possible that these SNPs modulate the function of p53 in maintaining the balance between tumor surveillance and stem cell regulation, which are important genetic modifiers for human longevity. Up till now, majority studies on p53 codon 72 SNP have focused on its impact upon cancer risk. However, there is no consensus in the literature as to the impact of p53 codon 72 SNP upon cancer risk (*van Heemst et al., 2005*; *Whibley et al., 2009*). Further, the role of p53 codon 72 SNP in stem cell regulation and aging process remains unclear. Several epidemiological studies of human populations indicate that p53 codon 72 SNP may influence human longevity. A perspective study with an aging human population ($\geq$85 years, n = 1226) reported that individuals homozygous for the P72 allele have a 41% increased survival (p=0.032) despite a 2.54-fold increased mortality from cancer (*van Heemst et al., 2005*). Another perspective study of the Danish general population (ages 20–95, n = 9219) reported that the p53 P72 allele is associated with an increased overall survival rate (*Bojesen and Nordestgaard, 2008*). Similarly, a study of long-lived individuals in Novosibirsk and Tyumen Regions (n = 131) reported the enrichment of the p53 P72 allele in the long-lived group (*Smetannikova et al., 2004*). These findings suggest that p53 activity is reversely associated with aging, and p53 codon 72 SNP may impact the lifespan in humans.

In this study, we used a genetic approach to investigate whether p53 codon 72 SNP modulates longevity through regulating the balance of p53 functions in tumor surveillance and stem cell regulation by using Hupki mice carrying p53 codon 72 SNP. To exclude the effect of mixed mouse genetic backgrounds on the lifespan and the aging process, mice carrying p53 codon 72 SNP were backcrossed with 129SV[sl] mice and C57BL/6J mice, respectively, for 10 generations. As shown in *Figure 1—figure supplement 1B and C* and *Table 1*, the lifespan and causes of death of these two mouse strains are very different. p53 codon 72 SNP showed a clear impact upon the lifespan in both strains of mice; P72 mice have a longer lifespan compared with their R72 littermates, although P72

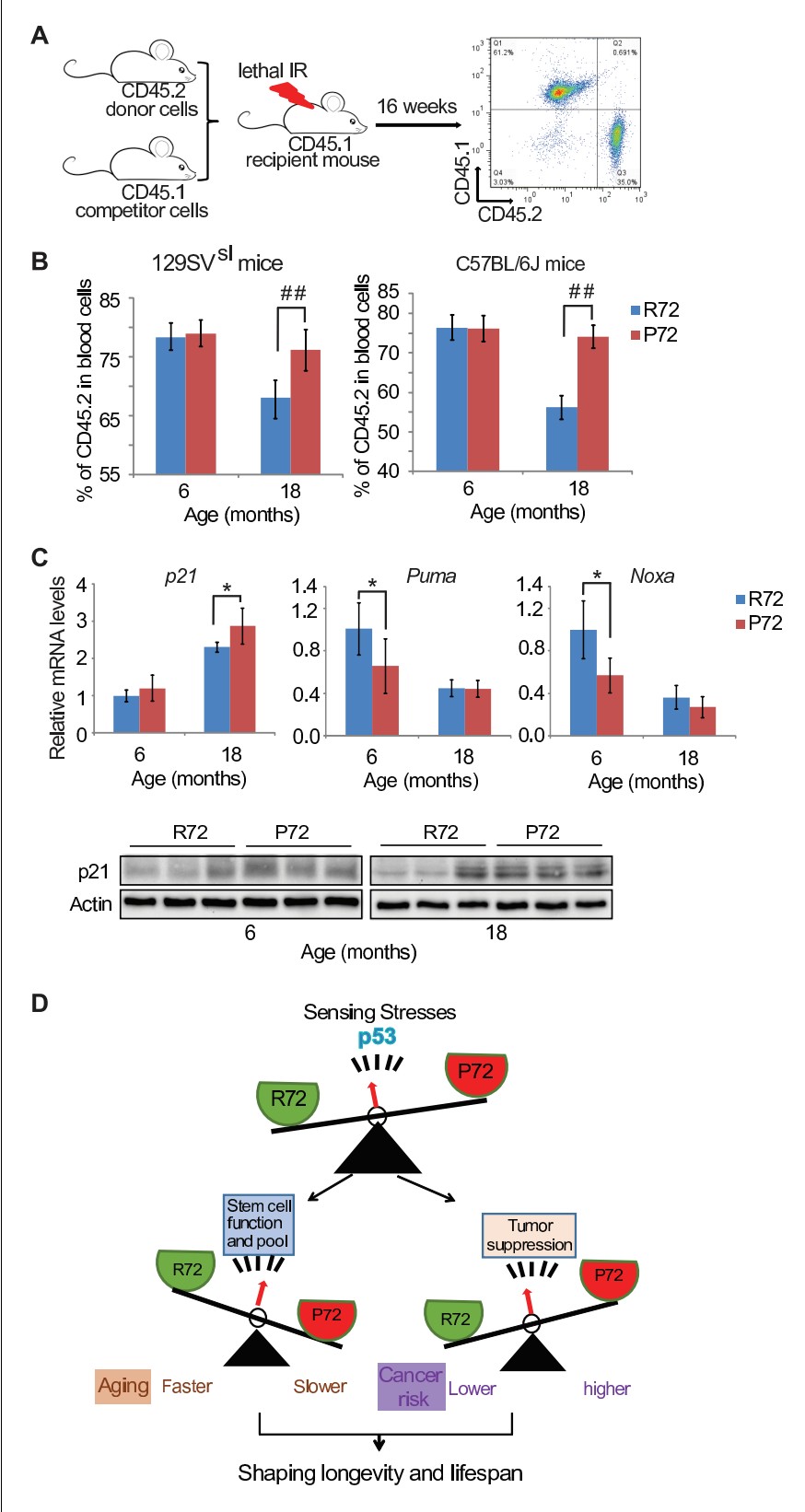

**Figure 6.** Aged p53 P72 mice exhibit better hematopoietic stem cell self-renewal and repopulation abilities compared with aged p53 R72 mice. (**A**) The experimental design for bone marrow transplantation assays to examine engraftment and repopulation abilities of HSCs in mice. (**B**) Percentage of
*Figure 6 continued on next page*

*Figure 6 continued*

CD45.2 cells in peripheral blood at 16 weeks after engraftment. In both 129SV^sl mice (left) and C57BL/6J mice (right), aged P72 mice showed better stem cell abilities of engraftment and repopulation compared with aged R72 mice. n = 6/group, ##: p<0.001; Student's *t*-test. (**C**) The mRNA expression levels of *p21*, *Puma* and *Noxa* (upper panels) and the protein expression levels of p21 (lower panels) in bone marrow from 129SV^sl p53 R72 and P72 mice at different ages as determined by real-time PCR assays and Western-blot assays, respectively. Data were presented as mean ± SD, n = 4/group, *p<0.05; Student's *t*-test. (**D**) A schematic model depicting the dual functions of p53 codon 72 SNP in regulating longevity.
DOI: https://doi.org/10.7554/eLife.34701.014
The following source data is available for figure 6:

**Source data 1.** Percentage of CD45.2 cells in peripheral blood at 16 weeks after engraftment.
DOI: https://doi.org/10.7554/eLife.34701.015

mice have a higher risk for tumor development. This result is consistent with the observation in human populations showing that P72 carriers have a longer lifespan (*van Heemst et al., 2005*; *Bojesen and Nordestgaard, 2008*; *Smetannikova et al., 2004*). Further, P72 mice display delayed aging-associated phenotypes compared with R72 mice. Results from this study further showed that P72 mice have a better self-renewal ability of stem cells and a delay of compensatory expansion of the stem cell pool compared with R72 mice. Stem cells from older P72 mice have better engraftment and repopulation ability compared with older R72 mice as determined by the bone marrow transplantation assays (*Figure 6D*). It has been suggested that p53 codon 72 SNP can influence the basal expression levels of some p53 target genes in humans, including *p21* and *PAI-1* (*Salvioli et al., 2005*, *Testa et al., 2009*). For instance, it was reported that dermal fibroblasts from P72 carriers display a higher expression of *p21* (*Salvioli et al., 2005*). In plasma samples from healthy populations, the P72 allele plays an important role in determining PAI-1 levels in aging populations (*Testa et al., 2009*). Results from this study showed that the bone marrow from P72 mice displays higher expression levels of *p21* but lower expression levels of *Puma* and *Noxa* compared with R72 mice. The differential expression of these target genes may modulate the p53 decision on cell fates towards survival or death, which may contribute to the delayed aging process in HSC function observed in P72 mice. In addition to p53 codon 72, a group of functional SNPs have been identified in the *p53* gene and important genes in the p53 pathway, such as *MDM2*. A very recent study reported that a patient affected by a segmental progeroid syndrome has a germline mutation in the *MDM2* gene (*Lessel et al., 2017*). This mutation abrogates MDM2 function and leads to increased p53 levels and function, which might be the driving cause for the premature aging phenotype of this patient (*Lessel et al., 2017*). It will be of interest to study how these SNPs in the p53 pathway (individual or in combination) impact aging and longevity in future studies.

Taken together, results from this study provided the genetic evidence showing that functional p53 codon 72 SNP, which regulates the activity of p53, influences aging and longevity through the regulation of self-renewal function of stem cells. Results from this study strongly support a role of p53 in regulation of stem/progenitor cell function and longevity.

## Materials and methods

### Mice

Hupki mice carrying either the P72 or R72 allele were generous gifts from Dr. Maureen Murphy (The Wistar Institute) (*Kung et al., 2016*). Hupki mice in 129SV^sl and C57BL/6J backgrounds were produced by backcrossing Hupki mice ten times to 129SV^sl and C57BL/6J, respectively. C57BL/6J CD45.1 mice (RRID:IMSR_JAX:002014) were purchased from The Jackson Laboratory (Bar Harbor, ME). All animal experiments were approved by the IACUC committee of Rutgers University.

### Micro-CT scan analysis

Mice were anesthetized for CT scanning of whole body skeletons using the INVEON PET/CT system (Siemens Healthcare). The images were reconstructed using INVEON Research Workplace software (Siemens Healthcare, Tarrytown, NY).

The bone microstructure measurement was carried out as previously described (*Ell et al., 2013*). In brief, mouse tibias were scanned by micro-CT. The images were reconstructed with Beam

Hardening Correction and Hounsfield calibrated before being analyzed using INVEON Research Workplace software. The 3D images were generated corresponding to the trabecular bone regions. CT scans were carried out at the Preclinical Imaging Shared Resource of Rutgers Cancer Institute of New Jersey.

## Histology

Paraffin-embedded skin specimens were sectioned with 5 µm thickness and stained with hematoxylin and eosin (H and E). The thickness of the dermal and adipose layers from the skin samples were determined by taking three random measurements along the length of each skin sample using ImageJ software.

## Cutaneous wound healing assays

Cutaneous wound healing assays were carried out as previously described (*Tyner et al., 2002*). In brief, mice were anesthetized and a full-thickness wound was generated in the mouse dorsal skin using a 3 mm biopsy punch (Integra, York, PA). Wound diameters were measured daily. Wound areas = $0.25 \times \pi \times$ width $\times$ length.

## Flow cytometry (FCM) assays to determine LT-HSC numbers and HSC proliferation

LT-HSC numbers were determined as previously described (*Dumble et al., 2007*). In brief, bone marrow cells were flushed out from mouse hind limb bones with PBS and stained with a cocktail of antibodies (BD bioscience Pharmingen), including an anti-lineage-APC antibody (RRID:AB_1645213), an anti-Sca-1-PE-Cy7 antibody (RRID:AB_647253), an anti-c-kit-PE-CF594 antibody (RRID:AB_11154233), an anti-CD34-FITC antibody (RRID:AB_395017) and an anti-Flk-2-PE antibody (RRID:AB_395079). LT-HSCs which were selected as Lin$^{-/low}$, Sca1$^+$, c-kit$^+$ and CD34$^-$, Flk2$^-$ cells were quantified by FCM analysis using a Beckman-Coulter Cytomics FC500 Flow Cytometer (Indianapolis, IN).

To determine the HSC proliferation ability, mice were injected intraperitoneally with 1 mg BrdU (BD bioscience Pharmingen) at 16 hr before the collection of bone marrow. Bone marrow cells were stained with a cocktail of antibodies, including an anti-lineage-APC antibody, an anti-Sca-1-PE-Cy7 antibody, an anti-c-kit-PE-CF594 antibody and an anti-Flk-2-PE antibody. After cell surface staining, a BrdU-FITC Flow Kit (BD bioscience Pharmingen; RRID:AB_2617060) was used to identify cycling cells according to the manufacturer's instructions. Proliferating HSCs were identified as Lin$^{-/low}$, Sca1$^+$, c-kit$^+$, Flk2$^-$ and BrdU$^+$ by FCM analysis.

## Total bone marrow competitive transplantation assays

Bone marrow transplantation assays were carried out as previously described (*Dumble et al., 2007*). In brief, bone marrow cells from 6-month-old and 18-month-old 'donor' CD45.2 mice were mixed with bone marrow cells from 6-month-old 'competitor' CD45.1 mice at a ratio of 2:1. Recipient CD45.1 mice at the age of 6 to 12 week-old were irradiated with a lethal dose of 10 Gy the day before bone marrow transplantation. n = 6/group. The mixture of bone marrow cells was injected into recipient mice *via* the tail vein. Sixteen weeks after transplantation, peripheral white blood cells of recipient mice were analyzed for CD45.1 and CD45.2 cell surface markers using an anti-CD45.1-PE antibody (RRID:AB_395044) and an anti-CD45.2-FITC antibody (RRID:AB_395041), respectively (BD Biosciences Pharmingen).

## Western-blot assays

Standard Western-blot assays were used to analyze protein expression in tissues. The following antibodies were used for assays: anti-p53 (FL393, Santa Cruz Biotechnology; RRID; AB_653753), anti-p21 (Santa Cruz Biotechnology; RRID:AB_628073), and β-actin (Sigma; RRID:AB_476744).

## Taqman real-time PCR

Total RNA was prepared by using an RNeasy kit (Qiagen). All probes were purchased from Applied Biosystems. Real-time PCR was done in triplicate with TaqMan PCR mixture (Applied Biosystems). The expression of genes was normalized to the *β-actin* gene.

## Statistical analysis

The data were present as mean ± SD. The lifespan of mice were summarized by Kaplan-Meier plots and compared using the log-rank test using GraphPad Prism software. All other *p* values were obtained using the Student's *t*-test. Based on survival data of p53 codon 72 SNP mice, we hypothesized that the P72 mice have a delayed development of aging associated phenotypes. Therefore, one-tailed Student's *t*-test was used for majority of data analysis related to the development of aging associated phenotypes. Values of $p < 0.05$ were considered to be significant.

## Acknowledgements

This study was supported by Lawrence Ellison Foundation, National Institutes of Health (NIH) Grants 1R01CA160558, 1R01CA203965 and DoD W81XWH-16-1-0358 (to WH), and 1R01CA227912 (to ZF and WH). YZ is supported by NIH F99CA222734. XY is supported by NJCCR Postdoctoral Fellowship Award. This research was supported by the Flow Cytometry/Cell Sorting shared resource of Rutgers Cancer Institute of New Jersey and the Preclinical Imaging shared resource of Rutgers Cancer Institute of New Jersey (P30CA072720).

## Additional information

### Funding

| Funder | Grant reference number | Author |
|---|---|---|
| National Institutes of Health | F99CA222734 | Yuhan Zhao |
| New Jersey Commission on Cancer Research | Postdoctoral Fellowship Award | Xuetian Yue |
| National Institutes of Health | 1R01CA227912 | Zhaohui Feng Wenwei Hu |
| Lawrence Ellison Foundation | New Investigate Award AG-NS-0781-11 | Wenwei Hu |
| National Institutes of Health | 1R01CA160558 | Wenwei Hu |
| National Institutes of Health | 1R01CA203965 | Wenwei Hu |
| U.S. Department of Defense | W81XWH-16-1-0358 | Wenwei Hu |

The funders had no role in study design, data collection and interpretation, or the decision to submit the work for publication.

### Author contributions

Yuhan Zhao, Data curation, Formal analysis, Investigation, Methodology, Writing—original draft; Lihua Wu, Data curation, Investigation, Methodology; Xuetian Yue, Cen Zhang, Jianming Wang, Data curation, Formal analysis, Investigation; Jun Li, Data curation; Xiaohui Sun, Yiming Zhu, Formal analysis; Zhaohui Feng, Formal analysis, Supervision, Funding acquisition, Investigation, Project administration, Writing—review and editing; Wenwei Hu, Conceptualization, Formal analysis, Supervision, Funding acquisition, Investigation, Project administration, Writing—review and editing

### Author ORCIDs

Yuhan Zhao (iD) http://orcid.org/0000-0001-5529-1907
Wenwei Hu (iD) http://orcid.org/0000-0003-3971-4257

### Ethics

Animal experimentation: This study was performed in strict accordance with the recommendations in the Guide for the Care and Use of Laboratory Animals of the National Institutes of Health. All of the animal experiments were approved by institutional animal care and use committee (IACUC) protocol (I14-012) of the University of Rutgers.

**Decision letter and Author response**
Decision letter https://doi.org/10.7554/eLife.34701.018
Author response https://doi.org/10.7554/eLife.34701.019

## Additional files

**Supplementary files**
• Transparent reporting form
DOI: https://doi.org/10.7554/eLife.34701.016

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
