## [Decision Letter]

Thank you for submitting your article "A polymorphism in the tumor suppressor p53 affects aging and longevity in mouse models" for consideration by *eLife*. Your article has been reviewed by three peer reviewers, and the evaluation has been overseen by a Reviewing Editor and James Manley as the Senior Editor. The following individual involved in review of your submission has agreed to reveal his identity: Phillip Buckhaults (Reviewer #2).

The reviewers have discussed the reviews with one another and the Reviewing Editor has drafted this decision to help you prepare a revised submission.

Summary:

This is a well-performed study supporting a role of a polymorphism in TP53 in the aging process. These data compliment existent data in humans suggesting that the Pro72 SNP of p53 confers longevity, and the authors have performed a detailed analysis in an inbred mouse model. Though the work is well done and likely to be of interest, all three Reviewers noted a failure of the authors to address mechanism; for example, an unbiased RNA Seq analysis of young versus old tissues or bone marrow cells could provide keys as to the underlying mechanism for the phenotypic differences. The authors are encouraged to include added experimentation in order to address potential mechanism.

In this manuscript the authors examine the potential role of the codon 72 polymorphism of p53 on aging and cancer risk in a mouse model, in two different inbred backgrounds. The authors find that the Pro72 SNP is associated with increased risk for cancer, but also increased longevity. They perform a number of well-done assays on aging phenotypes in the mouse, and again the Pro72 SNP is associated with increased aging phenotypes. In particular the data in Figure 6 showing that stem cells from Pro72 mice have better engraftment in a bone marrow transplantation assay are quite compelling that stem cell function may be influenced by the codon 72 SNP in p53.

Essential revisions:

1) The authors fail to uncover underlying mechanism for their findings that Pro72 mice live longer and/or have improved bone marrow engraftment. Ideally an unbiased RNA Seq analysis of old and young tissues or bone marrow cells from Pro72 and Arg72 mice could be performed and would lend excitement to the manuscript. Alternatively, there are some studies that find Pro72 over-represented in centenarians (see Salvioli et al., 2005; Testa et al., 2009) and these studies have suggested that p53 regulation of p21/waf1 or PAI-1 may be the underlying mechanism.

These manuscripts should be referenced, and the authors should make some experimental effort to address mechanism. The Discussion section should likewise include some discussion of the potential mechanism(s) uncovered by the authors.

2) There is a question about some of the statistical analyses done by the authors. As one example, Figure 2A, the median survival difference of 481 days and 495.5 days seems hardly significantly different and the survival curves overlap extensively, yet the authors find a p value of 0.0015 using the log rank test. Can the authors please revisit and explain this analysis, which does not seem accurate.

3) Similar to the concern above, the analysis of Lordokyphosis in Figure 3B shows large error bars that overlap and a very small sample size of n=5. Yet the authors indicate a p value under 0.05. How is this being measured? More details of their analysis from a bio-statistician should be included.

---

## [Author Response]

This is a well-performed study supporting a role of a polymorphism in TP53 in the aging process. These data compliment existent data in humans suggesting that the Pro72 SNP of p53 confers longevity, and the authors have performed a detailed analysis in an inbred mouse model. Though the work is well done and likely to be of interest, all three Reviewers noted a failure of the authors to address mechanism; for example, an unbiased RNA Seq analysis of young versus old tissues or bone marrow cells could provide keys as to the underlying mechanism for the phenotypic differences. The authors are encouraged to include added experimentation in order to address potential mechanism.In this manuscript the authors examine the potential role of the codon 72 polymorphism of p53 on aging and cancer risk in a mouse model, in two different inbred backgrounds. The authors find that the Pro72 SNP is associated with increased risk for cancer, but also increased longevity. They perform a number of well-done assays on aging phenotypes in the mouse, and again the Pro72 SNP is associated with increased aging phenotypes. In particular the data in Figure 6 showing that stem cells from Pro72 mice have better engraftment in a bone marrow transplantation assay are quite compelling that stem cell function may be influenced by the codon 72 SNP in p53.

Thank the reviewers for the nice comments.

Essential revisions:1) The authors fail to uncover underlying mechanism for their findings that Pro72 mice live longer and/or have improved bone marrow engraftment. Ideally an unbiased RNA Seq analysis of old and young tissues or bone marrow cells from Pro72 and Arg72 mice could be performed and would lend excitement to the manuscript. Alternatively, there are some studies that find Pro72 over-represented in centenarians (see Salvioli et al., 2005; Testa et al., 2009) and these studies have suggested that p53 regulation of p21/waf1 or PAI-1 may be the underlying mechanism.These manuscripts should be referenced, and the authors should make some experimental effort to address mechanism. The Discussion section should likewise include some discussion of the potential mechanism(s) uncovered by the authors.

We agree with the reviewers that the manuscript can be strengthened by addressing the potential underlying mechanism for the finding of this study showing that P72 mice liver longer and have improved bone marrow engraftment ability. It is a good suggestion to perform the RNA-seq analysis of bone marrow cells from old and young P72 and R72 mice. However, the facility that we routinely used for RNA-seq analysis is currently having a longer than usual time period for sample processing and data analysis. Further, the basal expression levels of many p53 target genes are low and it is possible that the modulation of p53 codon 72 SNP on some p53 target genes only leads to slight changes in their expression levels. Real-time PCR assays are more sensitive in detecting slight difference in expression of low abundant transcripts compared with RNA-seq analysis. Therefore, we performed real-time PCR analysis of some p53 target genes involved in cell cycle arrest (*p21*) and apoptosis (*Puma* and *Noxa*) in the bone marrow from young (6 months) and older (18 months) P72 and R72 129SV^sl^ mice as the reviewers suggested.

As shown in Figure 6C, *p21* mRNA expression levels in the bone marrow were slightly higher in P72 mice compared with R72 mice as determined by real-time PCR assays with this difference being more obvious in older mice than in young mice. This difference in p21 expression levels was confirmed at the protein level as determined by Western-blot assays (Figure 6C). In contrast, the bone marrow from P72 mice displayed slightly lower expression levels of *Puma* and *Noxa* than that from R72 mice (Figure 6C). These results demonstrate the differential regulation of the basal expression levels of *p21, Puma* and *Noxa* by p53 codon 72 SNP in the bone marrow, which may contribute to the delayed aging process in HSC number and function observed in P72 mice.

In addition to the above-mentioned genes, the expression levels of *PAI-1* were examined in the bone marrow as the reviewer suggested and no obvious difference was observed between the bone marrow from P72 and R72 mice (data not shown).

We have also referenced the two relevant publications that the reviewers pointed out in our revised manuscript.

2) There is a question about some of the statistical analyses done by the authors. As one example, Figure 2A, the median survival difference of 481 days and 495.5 days seems hardly significantly different and the survival curves overlap extensively, yet the authors find a p value of 0.0015 using the log rank test. Can the authors please revisit and explain this analysis, which does not seem accurate.

The reviewer raised a very good question. We re-analyzed the data with the help from two bio-statisticians (Prof. Yiming Zhu, and Ms. Xiaohui Sun as listed in our manuscript as co-authors). The survival analysis was performed by the Kaplan-Meier plot and compared by the log-rank test. The p value of the survival results in Figure 2A is 0.015 which shows statistical difference between P72 and R72 mice. We apologize for the mistake of putting the p value in Figure 2A as 0.0015 in our original manuscript. We have corrected this mistake.

The log-rank test which is the most popular method of comparing the survival of groups, takes the whole follow up period into account. This statistical test is a large-sample chi-square test which calculates the chi-square for each event time for each group and sums the results. The summed results for each group are added to derive the ultimate chi-square to compare the full curves of each group (Rich et al., 2010). The advantage of this analysis is that this analysis does not require us to know anything about the shape of the survival curve or the distribution of survival times (Bland and Altman, 2004). It is therefore possible for survival curves with some overlap to have statistical significant difference. For Figure 2A, while survival curves have some overlap and median survival data are very close, survival curves are different after 75% survival. In both Figure 1 and Figure 2, lines indicating 50% and 75% survival have been labelled in each panel of the survival curve.

3) Similar to the concern above, the analysis of Lordokyphosis in Figure 3B shows large error bars that overlap and a very small sample size of n=5. Yet the authors indicate a p value under 0.05. How is this being measured? More details of their analysis from a bio-statistician should be included.

As suggested, we consulted the two above-mentioned bio-statisticians to re-visit the experimental design and re-perform the statistical analysis of all data in the manuscript. Based on survival data of p53 codon 72 SNP mice, we hypothesized that the P72 mice have a delayed development of aging associated phenotypes. Therefore, one-tailed Student’s *t*-test was used for majority of data analysis presented in Figure 3,Figure 4, Figure 5 and Figure 6, including Figure 3B. For data presented in Figure 3, due to the technical requirement, including processing capacity for micro-CT scan analysis, studies were designed with the age and gender-matched mice used as paired sets to reduce incidental variation. This explains that the analysis of Lordokyphosis in Figure 3B found the p value under 0.05 when the sample size was 5/group and the data showed large SD (standard deviation). All analyses were performed using GraphPad Prism. All source data have been included in the manuscript with the description of statistical analysis. Source data have been updated with the format displaying 2 digits after decimal point; p values have been updated with the format displaying 3 digits after decimal point. These details have been included in the methods and the source data as suggested.